# Food Waste and Qualitative Evaluation of Menus in Public University Canteens—Challenges and Opportunities

**DOI:** 10.3390/foods10102325

**Published:** 2021-09-30

**Authors:** Cristina Aires, Cristina Saraiva, Maria Conceição Fontes, Daniel Moreira, Márcio Moura-Alves, Carla Gonçalves

**Affiliations:** 1Department of Veterinary Sciences, School of Agriculture and Veterinary Sciences, University of Trás-os-Montes and Alto Douro, 5000-801 Vila Real, Portugal; acristina_aires@hotmail.com (C.A.); crisarai@utad.pt (C.S.); mcfontes@utad.pt (M.C.F.); danielfsmoreira.283@gmail.com (D.M.); marcioalves_7@hotmail.com (M.M.-A.); 2CECAV—Centre for Studies in Animal and Veterinary Science, University of Trás-os-Montes and Alto Douro, 5000-801 Vila Real, Portugal; 3CITAB—Centre for the Research and Technology of Agro-Environmental and Biological Sciences, University of Trás-os-Montes and Alto Douro, 5000-801 Vila Real, Portugal; 4CIAFEL—Research Centre for Physical Activity, Health and Leisure, Faculty of Sports, University of Porto, 4200-450 Porto, Portugal

**Keywords:** food waste, menu quality, canteen

## Abstract

Background: This study aims to evaluate food waste and menu quality in two canteens (A and B) from a Portuguese public university in order to identify challenges and opportunities to improve the food service. Methods: Food waste included the analysis of two canteens over 5 consecutive days by selective aggregate weighing. A qualitative evaluation of a 5-week menu cycle related to lunches was performed through the Qualitative Evaluation of Menus (AQE-d) method. Results: Both menus have “satisfactory” evaluations and lower adequacy to the dietary guidelines in criteria A, which evaluates general items from the dish, and in criteria B, which evaluates meat, fish and eggs. The calculated mean of food waste in both canteens exceeded the acceptable limit of 10%, except for the vegetarian (7.5%) dish in canteen A. The biggest waste was found in the vegetarian dish (16.8%) in canteen A. In meat dishes the conduit presents more waste (17.0%) than in fish and vegetarian dishes. Among these, the vegetables were the most wasted (25.3% and 27.9%, respectively). Conclusion: This work presents some insights to future interventions in the direction of a healthier and more sustainable foodservice.

## 1. Introduction

The period of life that is lived in university, usually between the ages of 18 and 24, is characterized by major changes in lifestyle, accompanied by an increase in independence and freedom from parental control and from the limitations of an active working life. This life stage can be associated with unbalanced food intake and food insecurity, which can have a detrimental impact on physical and mental health, on students’ cognitive, intellectual, and academic performance, and on future eating habits and health [1,2]. Therefore, it is recognized that universities need to develop public policies that promote healthy eating habits among students, such as interventions that promote healthy eating behaviors and improve access to healthy foods in the university environment, thereby facilitating the construction of eating patterns that will follow throughout adulthood [3]. Another aspect of relevance and growing importance is the need to actively face the challenges of climate change, with universities having to reduce the environmental impact of their campus [4].

Food waste imposes nutritional, economic, social, and environmental impacts on both developing and developed countries [5,6,7]. Reducing food loss and waste is widely seen as an important way to lower production costs, increase food system efficiency, improve food and nutrition security, and contribute to environmental sustainability [8].

Several studies have previously shown an unacceptable plate wastes index in university canteens [9,10] and conclude that this issue is caused by multidimensional factors, including individual-level characteristics and catering features [11]. Many higher education institutions have developed policies and strategies to reduce food waste [12].

The storage of food in inadequate conditions, the lack of meal planning that leads to the acquisition or preparation of excessive amounts of food, and the lack of awareness and knowledge of cookers, determine the large amounts of wasted food, both in households and in food services [13,14]. 

In canteens, three components of food waste can be observed, resulting from different stages of meal production [11]: waste arising from the storage and preparation stages; food prepared but not served, resulting from inadequate planning of the necessary quantities, which are commonly called leftovers [15]; and the quantities of food that are served but not consumed (plate waste) [16].

The definition of an acceptable level for food waste in collective feeding services does not reach consensus among the various authors and varies according to the characteristics of the unit and population for which it is intended. Strategies for reducing waste and the value considered acceptable should be based on values calculated in the feeding unit itself [17]. Rates of less than 10% were identified as acceptable for the relationship between the amount of food offered and that rejected by the consumer [18]. This amount is also accepted for collective food services [15]. The relevance and challenges of the study of wasted food in the collective food sector were recently recognized by the United Nations in the 2021 Food Waste Report [19]. 

To ensure the quality of service and avoid food waste, it is important that processes and services are standardized through the preparation of routines and operational technical procedures, as well as the planning of meals to be served by qualified professionals [20]. When preparing menus, it is important to consider the individual quantity of food per user, called per capita serving—a value that aims to ensure the balance of meals, obtain the quantities of raw material necessary for the confection, and acts as a parameter in controlling costs, production, and waste reduction [21]. 

Providing healthy meals in schools is one of the measures instilled by the World Health Organization to combat obesity [22]. School canteens assume an indispensable value and an increased responsibility, not only from a nutritional and food perspective about the composition of meals, but also in the process of socialization and the valorization of gastronomy. Canteens should play a role in including healthy eating habits, and where healthy eating habits should be taught, stimulated, and practiced [23]. 

Souza et al. conducted an intervention on the nutritional and sensory quality of the menus from children’s canteens and show the importance of adjustments in recipes and portioning of preparations to meet the nutritional recommendations and to control food waste [24]. The development of menus is complex because several aspects should be considered, such as: the nutritional needs of consumers, the habits and preferences of the population for which it is intended, the frequency of consumption, the turnover of consumers, the costs, sensory aspects, and the availability of raw materials, equipment, space, and employees [25]. 

The evaluation of the quality of the menus can be conducted in quantitative and qualitative terms [26]. The qualitative evaluation of the foods included in the menus, and their culinary methods, allows us to know whether these meals comply with the recommendations for a healthy diet [27]. If dietary recommendations are met, nutritional needs will naturally be met [28]. 

The aim of this study is to evaluate food waste and menu quality in two canteens from a Portuguese public university to identify challenges and opportunities to improve the food service.

## 2. Materials and Methods

### 2.1. Study Design

This transversal study was performed in two canteens (canteen A and canteen B) from one public university in northern Portugal during lunch service (between 12:00 and 14:30) during the second school semester. These canteens served lunch and dinners to students and workers from the university. One is located on the residences and the other on the university campus. The complete meal consists of soup, a meat, fish or vegetarian dish, dessert, and bread. Each day’s menu presents one option of either meat, fish or a vegetarian dish, which includes the conduit as the main source of protein (meat, fish, or eggs group, and/or pulse group, when applicable), the garnish is the main source of carbohydrates (group of cereals and its derivatives, tubers, and/or group of pulses, when applicable), and finally, vegetables are the main source of dietary fiber and micronutrients (group of vegetables). The consumers can choose to buy a complete meal (including soup, a meat, fish, or vegetarian dish, dessert, and bread) or only the dish at a lower price.

### 2.2. Qualitative Evaluation of Menus

Using the Method of Qualitative Evaluation of Menus (AQE), more specifically, the detailed tool (AQE-d) [25], an evaluation was made of a 5-week menu cycle related to lunches at the two canteens (May and June implemented menus). The AQE-d method evaluates the menu according to the accomplishment of 36 parameters, distributed according to the following evaluation criteria: A—General items of the complete plate (6 parameters evaluated); B—Meat, fish and egg (9 parameters evaluated); C—Cereals, derivatives, and tubers (3 parameters evaluated); D—Vegetables and pulses (5 parameters evaluated); E—Soup (6 parameters evaluated) and F—Desserts (5 parameters evaluated). The global classification of a menu is obtained by adding the values of the weighting of the criteria met and the percentage classification is obtained by the following formula:Percentage rating (%) = total score/68 × 100

The final classification of a menu is obtained by adding the weighting values of the met criteria. To allow for a qualitative classification of the menus, the percentage values had a correspondent classification: “very good”—≥90%, “good”—between ≥75% and <90%, “acceptable”—between ≥50% and <75%, and “not acceptable”, with a percentage of less than 50%.

### 2.3. Food Waste

The evaluation of food waste included the analysis of the two canteens that serve the academic community daily. In each canteen, the evaluation was performed over five consecutive days, between 18 May and 2 June 2021.

Food waste was assessed through selective aggregate weighing. Food weight was measured in grams by a high-precision portable electronic kitchen scale (Ruby^®^ model Delta) with a capacity of 15 kg and accuracy of 5 g. The total amount of food produced, leftovers, and scraps (plate waste) was weighed daily. The method used in weighing presupposes weighing the aggregated food by type of food, before the distribution of meals and after the meal, and after removing all non-edible residues. The same investigators carried out the measurements every day. Food remains from non-eating by all individuals were separated into different containers according to the type of food.

Measurements were taken in two stages each day, that is, the amount of food served and the amount of food scraps to obtain FW, using the following formula:Food Waste (%) = Food Scraps/Food Served × 100

### 2.4. Statistical Analysis

The analyses were carried out using the statistical software IBM SPSS STATISTICS Version 26. The statistical analysis involved measures of descriptive statistics (absolute and relative frequencies, means, and respective standard deviations). To compare means from independent samples, non-parametric tests were used—that is the Mann–Whitney test (to compare canteen A and B) and Kruskal–Wallis test (to compare types of dishes). The null hypothesis was rejected when the critical significance level was less than 0.05.

## 3. Results

Table 1 shows the qualitative evaluation of the menus. Both menus have an “acceptable” evaluation and lower adequacy (in %) was found in Criteria A and Criteria B.

Calculated mean of food waste, expressed as percentage of plate served (Table 2), in both canteens exceeded the acceptable limit of 10%, except for vegetarian dishes in canteen A. No significant differences were found in food waste between canteen A and canteen B (soup, *p* = 0.754; meat, *p* = 0.465; fish, *p* = 0.602; vegetarian, *p* = 0.347).

With regard to the dish components, it was found that in meat dishes the conduit present more waste (17.0%, *p* = 0.008), however in fish and vegetarian dishes no significant differences between dish components were found (*p* = 0.056) (Table 3).

## 4. Discussion

In relation to menu quality, criteria A, which evaluates general items from the dish, and criteria B, which evaluates meat, fish, and eggs, have the lowest accomplishment with the dietary guidelines. 

In criteria A, both menus showed the presence of dishes repeated in more than 5% of the meals, the use of foods rich in trans fats, and the monotony of cooking methods (during the 5 weeks evaluated). As positive aspects in this criterion highlights the lowest presence of fried products (maximum in 10% of the dishes) and the presence of at least two colors on the dish in all evaluated menus. According to other authors, the offer of repetitive dishes can be one of the causes of food waste [29]. On the other hand, the variety of colors of the foods provided on the menu is a crucial factor in meal planning, as this favors the varied supply of nutrients to customers. In addition, it arouses their interest, due to the presence of vibrant and contrasting colors — since the first customer–food contact is visual. Thus, preparations with varied colors provide greater acceptance by the consumer [30]. One study in university canteens in Brazil observed no good percentages for any region, with white and light yellow being prevalent. For example, a planned dish containing a cooked egg as the protein, a sautéed potato in the garnish, a dessert melon in the same menu, and lemon juice to drink [31].

In criteria B, both menus showed negative aspects such as the presence of red meat in 15% or more of the dishes, the absence of eggs as the main source of protein in 5 to 10% of dishes, and the absence of cooked meat free of skin and fat. Another aspect that affected the quality of the menus was the inequitable distribution of carbohydrate accompaniments (criteria C), or their occasional replacement by pulses. In addition, in most cases, the cooking method was the same.

According to the World Health Organization, it is recommended to consume 400 g of fruits and vegetables daily [30,32]. In this work, the daily supply of vegetables was verified, which is a positive point. The inexistence of additives to the horticultural accompaniment, such as flour, cheese, or bacon, also contributes to the quality of the menu. However, the supply of vegetables should be more varied and in accordance with the season. 

The criteria E (soup) and F (dessert) showed the highest acceptability rates. Among the parameters that contributed to the increased quality in soup were the daily supply of soup, the presence of vegetables in 90% of the soups, and the supply of pulses in at least one-fifth of the soups. Inversely, one study in collective catering showed insufficient supply of vegetables in soups [33]. In relation to the desserts, the positive aspects are the daily presence of fruit and the presence of desserts less than 7% of the time (only in canteen A). Benvindo et al. showed that the university menus evaluated did not present a varied offering of fruits every day [31].

It should be assumed that if the food is well prepared, the waste should be very close to zero [34], and some authors argue that food waste directly reflects the quality of meals and consumer satisfaction [35]. In a study in which questionnaires were applied to study the reasons for user waste in a canteen, concluded that the quality of food and the quantity served are determining factors of waste [36]. 

Thus, dishes that have high waste values should be rethought or changed during the menu planning. The comparative analysis between menu and the food waste can provide an answer in the fight against waste [37]. 

In this work was found that between 63.4% and 94.1% of the food produced is served with a very high standard deviation. The organization of this catering service does not have a pre-reservation system of the meals, so every day the number of consumers can have a huge variation, which explains the high standard deviation and makes it more difficult to produce food more precisely for the number of consumers (huge daily variations of the number of consumers). The food that was not served to the consumers is distributed to aid organizations of the region, and, for this reason, was not considered waste, but rather, increased the sustainability (environmental and social) of the food service [38]. In this specific case, the leftovers are tightly monitored regarding storage time and temperature, avoiding microbial growth until consumption, and the occurrence of diseases [39]. 

The proportion of food waste (plate waste) was high (above 10%) in almost all types of meals in both canteens (except in the vegetarian dish in canteen A). Waste values greater than 15% represent an indicator of poor service performance and less than 5% are classified as optimal [40]. In this study, no optimal value was found since the lowest calculated food waste was 7.5% (in the vegetarian dish in canteen A). One food service canteen reported 8.4% in total food waste [29] and, in another study in the restaurant of the University of Rio de Janeiro, all samples showed food waste values greater than 10% [41]. In Finnish student canteens a total food waste of 25.3% was found [42]. 

Contrary to our data, in other similar studies higher food waste values were observed when the protein component of the dish was fish and lower in the dishes with a meat protein component [9]. 

Soup waste also exceeds the acceptable level of waste in both canteens (11.5% in canteen A and 11.6% in canteen B) and presents higher values than similar studies (8.6% [10]). The dish components with an index of waste more than 20% was vegetables (in fish and vegetarian dishes) and the components that obtained an acceptable waste were the conduit from the vegetarian dish (2.3%). Comparing these results with other work in the same context, the conduit component was the most wasted (19.8%), and vegetables had a much lower wastage value than that obtained in this study (11.9%) [37]. The leftovers are often the healthiest plates and the vegetables in studies with young school aged children [43].

This work highlights the necessity to adopt strategies to reduce unnecessary plate waste. Different approaches have been studied before. Martin-Rios et al. [44] indicated that the top three drivers for adopting waste management initiatives are favorable cost-analysis, experimentation with existing management practices, and change in the existing business model. 

In our study, the option given to consumers to choose between a complete meal or only a dish, according to their appetite and with different pricing, could be one procedure friendly to reduce plate waste, as other authors pointed out previously [44]. However, the portions served in studied canteens are usually standardized, discouraging consumers from adapting their order, and this could be another way to reduce waste and provide a more personalized service [45]. The intervention with chefs and food handlers should be to give them more knowledge and training to reduce food waste integrated in different work routines, instead of educating professionals by providing guidelines [46]. The pre-ordering system in university canteens has not shown encouraging results with respect to food waste reduction [47].

Another measure could be the implementation of ‘doggy bag’ offers, to take away whatever is left on plates at the end of the meal [44]. Educating consumers to not throw away food and to have a conscientious consumption behavior, through information campaigns and mass media awareness, are common measures to change behaviors and lead to food waste prevention [38]. Ellison et al. tested the efficacy of a food waste reduction campaign in a university dining hall and showed a non-significant impact on waste behavior, but significant changes in students’ beliefs related to food waste [48], which could be an important step forward in foodservice sustainability [49]. Pinto et al. implemented one education campaign in university students, which showed around 15% of plate waste [10]. 

The implementation of one systematic monitoring system of food waste produced in catering units could also be challenging [50], so automatic technology for quantifying food waste could be important for future wastage quantification [51].

One limitation of our study could be that the number of meals served per day could have been lower than the period before the COVID-19 pandemic, because fewer students were having lunch on campus. A strength of the present study could be identifying the direct methodology adopted for measuring food waste.

## 5. Conclusions

This work presents robust data about food waste and menu quality that highlight the challenges and opportunities to change in order to have a healthier and more sustainable foodservice.

The evaluation of the menus of both canteens was “acceptable”. Healthier cooking methods should continue to be chosen, prioritizing the consumption of soup, vegetables, pulses, and fruits, particularly in their season of production. In addition, during the preparation of menus, it is important to consider the nutritional and sensory aspects, and the prioritizing of fresh and minimally processed foods, since the meals offered to students are vehicles to promote adequate and healthy food.

Regarding the assessment of food waste in canteens, the results showed that food waste is far above the limits that are considered acceptable. It is necessary to adopt strategies to reduce unnecessary waste with raw materials, for example, through the implementation of new menus, training of workers, giving the consumer opportunities to choose the desired amount (portion served), develop food satisfaction/food preference questionnaires, and carrying out campaigns to raise awareness about food waste. Carrying out initiatives such as these will certainly contribute to improving the social, economic, and environmental sustainability of the university.

## Figures and Tables

**Table 1 foods-10-02325-t001:** Qualitative analysis of the lunch menu served in canteens (A and B) from public university (% of adequacy).

Canteen	Menu Weeks	Criteria A	Criteria B	Criteria C	Criteria D	Criteria E	Criteria F	Global Adequacy
A	5	33.3	38.9	50.0	50.0	66.7	75.0	52.9
B	5	33.3	33.3	66.7	50.0	66.7	58.3	50.0

Criteria A—General items of the complete dish; Criteria B—Meat, fish, and egg; Criteria C—Cereals, derivatives, and tubers; Criteria D—Vegetables and pulses; Criteria E—Soup; Criteria F—Dessert.

**Table 2 foods-10-02325-t002:** Food waste evaluated during 5 weekdays during lunch served in canteens (A and B) from public university.

	Meals Served *n*	Food Produced	Food Served		Food Wasted	
		g	g	% ^a^	g	% ^b^
**Canteen A**						
Soup		19,513.7 ± 2914.6	14,255.8 ± 2807.8	74.0 ± 14.0	1583.1 ± 780.7	11.5 ± 6.2
Meat dish	358	38,946.0 ± 7761.9	34,216.0 ± 7127.8	88.0 ± 6.5	4406.6 ± 2009.5	14.3 ± 7.8
Fish dish	59	8718.2 ± 2226.3	7561.6 ± 2046.7	87.1 ±13.3	688.2 ± 211.1	10.4 ± 6.3
Vegetarian dish	35	6722.4 ± 2310.5	5540.2 ± 3277.3	75.0 ± 29.9	381.1 ± 245.9	7.5 ± 3.0
**Canteen B**						
Soup		56,201.0 ± 5679.1	35,741.0 ± 10674.3	63.4 ± 15.9	4124.8 ± 1455.1	11.6 ± 4.8
Meat dish	795	64,172.0 ± 12,579.8	48,740.6 ± 19.1	78.4 ± 19.1	5682.6 ± 2258.2	12.1 ± 5.3
Fish dish	274	29,877.3 ± 7695.1	27,648.2 ± 5313.1	94.1 ± 8.1	2933.9 ± 1058.7	10.3 ± 2.4
Vegetarian dish	100	16,459.3 ± 8333.3	14,973.9 ± 7571.4	90.3 ± 7.7	1958.2 ± 1390.0	16.8 ± 11.9

*n*—number of meals served, ^a^—% from food produced that was served to consumers; ^b^—% from food served to consumers that was wasted.

**Table 3 foods-10-02325-t003:** Food waste (%) characterization in canteens (A and B) from public university.

		N ^a^	Conduit ^b^	Garnish ^b^	Vegetables ^b^
Meat dish	Total	8	17.0 ± 8.5	14.0 ± 5.2	11.3 ± 8.1
	Canteen A	4	23.6 ± 5.4	15.3 ± 2.9	8.1 ± 4.8
	Canteen B	4	10.4 ± 5.1	12.7 ± 6.5	14.4 ± 9.4
Fish dish	Total	7	10.7 ± 8.0	13.4 ± 6.3	25.3 ± 16.4
	Canteen A	4	10.6 ± 8.8	14.6 ± 7.7	31.9 ± 18.6
	Canteen B	3	10.9 ± 6.7	11.7 ± 2.8	16.5 ± 5.8
Vegetarian dish	Total	5	2.3 ± 1.2	11.6 ± 4.1	27.9 ± 20.7
	Canteen A	3	3.1 ± 1.0	11.4 ± 5.2	34.5 ± 23.9
	Canteen B	2	10.9 ± 6.7	11.7 ± 2.8	16.5 ± 5.8

^a^—number of days analyzed (excluding days with compound dishes); ^b^—food waste expressed in % mean ± dp.

## Data Availability

The raw data supporting the results could be shared by the authors after inquiry directed to the corresponding author.

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
