# Peer review of "Food Waste and Qualitative Evaluation of Menus in Public University Canteens—Challenges and Opportunities"

_foods, 2021, doi:10.3390/foods10102325_

Round 1

Reviewer 1 Report

I’m very supportive of this paper’s aims and appreciative of the clear writing in much of the paper. I agree that more clarity and cohesion around the notion of food waste in universities is sorely needed. This paper has many strengths and some opportunities for improvement, both of which I will elaborate below. 

  • Your research objective reads too descriptive, "evaluate food waste and menu quality".  I suggest authors try to dig down deeper into existing literature and research to strengthen their contribution. For example, a recent paper published by Martin-Rios et al. (2018) addresses wastage from the innovation perspective. Hennchen (2019) applies practice theory. Martin-Rios et al., (2021) technological innovations, and Messner et al. (2020) paradox theory. Authors should include that stream of research early on their study. Also, some recent work has explored the relevance of different diversion practices for wastage mitigation in canteens and dining halls (references below). Some of these works (e.g. Ellison et al., 2020) should also be cited early in the introduction.
  • Just as the theoretical model is not fully developed in the paper, there is no reference to hypotheses, nothing is said on this point, which makes the paper appear unreflective. 
  • Discussion and Conclusions: This section seems to need a more clear structure to present its ideas and discoveries. Some links seem rather tangentially linked, such as the lack of conclusions derived from your robust data. The flow in this section does not build toward an emerging clarity about what you discovered.  Can you step back and think – what is it I am trying to convey. Step back and consider what might be done with this section.

References

Ellison, B., Savchenko, O., Nikolaus, C. J., & Duff, B. R. (2019). Every plate counts: Evaluation of a food waste reduction campaign in a university dining hall. Resources, Conservation and Recycling, 144, 276-284.

Hennchen, B. (2019). Knowing the kitchen: Applying practice theory to issues of food waste in the food service sector. Journal of Cleaner Production, 225, 675-683.

Martin-Rios, C., Hofmann, A., & Mackenzie, N. (2021). Sustainability-oriented innovations in food waste management technology. Sustainability, 13(1), 210.

Martin-Rios, C., Demen-Meier, C., S. Gossling, C. Cornuz (2018) Food waste management innovations in the foodservice industry, Waste Management, 79, pp. 196-206

Messner, R., Richards, C., & Johnson, H. (2020). The “Prevention Paradox”: food waste prevention and the quandary of systemic surplus production. Agriculture and Human Values, 1-13.

Pinto, R. S., dos Santos Pinto, R. M., Melo, F. F. S., Campos, S. S., & Cordovil, C. M. D. S. (2018). A simple awareness campaign to promote food waste reduction in a University canteen. Waste management, 76, 28-38. 

Author Response

  1. DETAILED RESPONSE TO REVIEWER 1

Reviewer #1: I’m very supportive of this paper’s aims and appreciative of the clear writing in much of the paper. I agree that more clarity and cohesion around the notion of food waste in universities is sorely needed. This paper has many strengths and some opportunities for improvement, both of which I will elaborate below.

Your research objective reads too descriptive, "evaluate food waste and menu quality".  I suggest authors try to dig down deeper into existing literature and research to strengthen their contribution. For example, a recent paper published by Martin-Rios et al. (2018) addresses wastage from the innovation perspective. Hennchen (2019) applies practice theory. Martin-Rios et al., (2021) technological innovations, and Messner et al. (2020) paradox theory. Authors should include that stream of research early on their study. Also, some recent work has explored the relevance of different diversion practices for wastage mitigation in canteens and dining halls (references below). Some of these works (e.g. Ellison et al., 2020) should also be cited early in the introduction. Just as the theoretical model is not fully developed in the paper, there is no reference to hypotheses, nothing is said on this point, which makes the paper appear unreflective.

Our reply: We considerer your recommendation. The suggested authors and works were included in discussion section and introduction in order to strengthen the work – lines 228 to 260.

Reviewer #1: Discussion and Conclusions: This section seems to need a more clear structure to present its ideas and discoveries. Some links seem rather tangentially linked, such as the lack of conclusions derived from your robust data. The flow in this section does not build toward an emerging clarity about what you discovered.  Can you step back and think – what is it I am trying to convey. Step back and consider what might be done with this section.

 Our reply: The authors sincerely thanks to reviewer and the opportunity to strengthen the work – lines 228 to 260.

Reviewer 2 Report

General comment

This study presents some data on the food waste produced in two university canteens. The topic is relevant and the results are interesting. The manuscript is well written. However, some sections could be improved.  I have some specific concerns (reported below).

Specific comments

L87. The Study design description misses relevant information. I would suggest providing more details regarding the criteria used to select the two canteens and about the characteristics of the two canteens in terms of location (cities) and size (number of people served in average every day), meals analysed (lunches, dinners, both?), season of the year of data collection, payment system of the meals (one price for a complete menu or a price for each dish?). All these variables could have an effect on the produced food waste (e.g. if people pay a predefined quantity of money for a multiple dish meal the probability to have food waste is higher than if people pay for each dish they order). Thus, it is important to report the details and discuss the potential effect of those variables.

L107. It would be interesting to read more details on the food waste method, particularly related to the staff in charge of measuring. How many people were in charge of aggregating and weighting food? Different people in different days would operatively act in a different way. Thus, did you avoid this bias training the staff? How?

Table 2. Numbers for Soup seem missed. Please add them.

L169. The Discussion section presents some results note reported in the previous section. I would suggest enriching the results section with some data regarding the evaluation of the menus.

L174. “… (during the 5 weeks evaluated)…”. 5 weeks or 5 days? Please check and revise.

L229. Please substitute 8.39% with 8.4% (to be consistent with the format of the other cited % values).

L237. Please substitute 8.60% with 8.6% (to be consistent with the format of the other cited % values).

I would suggest citing (in the introduction/discussion) also the following recent papers reporting new knowledge on food waste in the context of school/university canteens:

Migliavada, R.; Ricci, F.Z.; Torri, L. A three-year longitudinal study on the use of pre-ordering in a university canteen. Appetite 2021, 163, 105203, doi:10.1016/j.appet.2021.105203.

Liz Martins, M.; Rodrigues, S.S.P.; Cunha, L.M.; Rocha, A. Factors influencing food waste during lunch of fourth-grade school children. Waste Manag. 2020, 113, 439–446, doi:10.1016/j.wasman.2020.06.023.

Lagorio, A.; Pinto, R.; Golini, R. Food waste reduction in school canteens: Evidence from an Italian case. J. Clean. Prod. 2018, 199, 77–84, doi:10.1016/j.jclepro.2018.07.077.

Kasavan, S.; Ali, N.I.B.M.; Ali, S.S.B.S.; Masarudin, N.A.B.; Yusoff, S.B. Quantification of food waste in school canteens: A mass flow analysis. Resour. Conserv. Recycl. 2021, 164, 105176, doi:10.1016/j.resconrec.2020.105176.

García-Herrero, L.; De Menna, F.; Vittuari, M. Food waste at school. The environmental and cost impact of a canteen meal. Waste Manag. 2019, 100, 249–258, doi:10.1016/j.wasman.2019.09.027.

García-Herrero, L.; Costello, C.; De Menna, F.; Schreiber, L.; Vittuari, M. Eating away at sustainability. Food consumption and waste patterns in a US school canteen. J. Clean. Prod. 2021, 279, 123571, doi:10.1016/j.jclepro.2020.123571.

Derqui, B.; Grimaldi, D.; Fernandez, V. Building and managing sustainable schools: The case of food waste. J. Clean. Prod. 2020, 243, doi:10.1016/j.jclepro.2019.118533.

Derqui, B., Fernandez, V., & Fayos, T. (2018). Towards more sustainable food systems. Addressing food waste at school canteens. Appetite, 129, 1–11. https://doi. org/10.1016/j.appet.2018.06.02.

Ali, A.Y.; Ayele, A. Contribution of quality tools for reducing food waste in university canteen. J. Appl. Res. Ind. Eng. 2019, doi:10.22105/JARIE.2019.177566.1086.

Author Response

  1. DETAILED RESPONSE TO REVIEWER 2

Reviewer #2: General comment

This study presents some data on the food waste produced in two university canteens. The topic is relevant and the results are interesting. The manuscript is well written. However, some sections could be improved.  I have some specific concerns (reported below).

Our reply: The authors sincerely thanks the comments from the reviewer.

Reviewer #2: L87. The Study design description misses relevant information. I would suggest providing more details regarding the criteria used to select the two canteens and about the characteristics of the two canteens in terms of location (cities) and size (number of people served in average every day), meals analysed (lunches, dinners, both?), season of the year of data collection, payment system of the meals (one price for a complete menu or a price for each dish?). All these variables could have an effect on the produced food waste (e.g. if people pay a predefined quantity of money for a multiple dish meal the probability to have food waste is higher than if people pay for each dish they order). Thus, it is important to report the details and discuss the potential effect of those variables.

Our reply: The authors appreciate the opportunity to clarify these information’s on methods section – lines: 87-95, 236-244.

Reviewer #2: L107. It would be interesting to read more details on the food waste method, particularly related to the staff in charge of measuring. How many people were in charge of aggregating and weighting food? Different people in different days would operatively act in a different way. Thus, did you avoid this bias training the staff? How?

Our reply: The authors appreciate the opportunity to clarify these information’s on methods section – line: 116.

Reviewer #2: Table 2. Numbers for Soup seem missed. Please add them.

Our reply: Thanks for the comment. It was not possible to quantify the number of soups, as the meals were only registered in the payment box as a complete meal, however, the soup accompaniment was optional and not all customers took soup. So, it was not possible to determine precisely how many soups were consumed. However, we were able to accurately estimate the amount of soup produced, served and wasted (plate wasted).

Reviewer #2: L169. The Discussion section presents some results note reported in the previous section. I would suggest enriching the results section with some data regarding the evaluation of the menus.

Our reply: Thanks for the comment. We strength the discussion section.

Reviewer #2: L174. “… (during the 5 weeks evaluated)…”. 5 weeks or 5 days? Please check and revise.

Our reply: Thanks for the comment. Indeed the menu evaluation was performed for the menus that are implemented in canteens during the data collection. These menus are correspondent to a 5-week menu cycle – line 98.

Reviewer #2: L229. Please substitute 8.39% with 8.4% (to be consistent with the format of the other cited % values).

Our reply: Amended.

Reviewer #2: L237. Please substitute 8.60% with 8.6% (to be consistent with the format of the other cited % values).

Our reply: Amended.

Reviewer #2: I would suggest citing (in the introduction/discussion) also the following recent papers reporting new knowledge on food waste in the context of school/university canteens

Our reply: The authors sincerely thanks to reviewer and the opportunity to strengthen the work with the recent papers suggested – lines 228 to 260.

Round 2

Reviewer 2 Report

The Authors well improved the manuscript taking into account all my comments. I do not have any other concerns.